# Changes in background characteristics and risk factors among SIDS infants in England: cohort comparisons from 1993 to 2020

Anna Pease ,[1] Nicholas Turner ,[1] Jenny Ingram ,[1] Peter Fleming,[1] Karen Patrick,[2] Tom Williams,[3] Vicky Sleap,[3] Kieren Pitts ,[4] Karen Luyt ,[3] Becky Ali ,[1] Peter Blair [1]

[1]Population Health Sciences, University of Bristol Medical School, Bristol, UK
[2]Research and Development, Royal United Hospitals Bath NHS Foundation Trust, Bath, UK
[3]Translational Health Sciences, University of Bristol Medical School, Bristol, UK
[4]Research IT, University of Bristol, Bristol, UK

**Correspondence to**
Dr Anna Pease;
a.pease@bristol.ac.uk

## ABSTRACT

**Objectives** Using the National Child Mortality Database, this work aims to investigate background characteristics and risk factors in the sleeping environment associated with sudden infant death syndrome (SIDS) and compare the prevalence with previous English SIDS case–control studies.

**Design** Cohort of SIDS in 2020 compared with a combined analysis of two case–control studies conducted in 1993–1996 and 2003–2006.

**Setting** England, UK

**Participants** 138 SIDS deaths in 2020 compared with 402 SIDS deaths and 1387 age-equivalent surviving controls, combined from previous studies.

**Results** The increased vulnerability of SIDS infants identified in previous studies has become more marked. The infants who died in 2020 were younger (median=66 days (IQR: 34–118) vs 86 days (IQR: 52–148), p=0.003) with an increased prevalence of low birth weight (30.5% vs 21.6%, p=0.04) and preterm births (29.6% vs 19.3%, p=0.012). The excess of socioeconomically deprived families, male infants and high levels of maternal smoking during pregnancy were still evident. Among recent deaths, fewer infants were put down or found on their side; however, there was no significant change in the proportion of infants who were put down (15.6% vs 14.6%, p=0.81) and found prone (40.4% vs 35.3%, p=0.37), despite population wide risk reduction advice over three decades. The proportional increase observed in 2003–2006 of half the deaths occurring while sleeping next to an adult was maintained in 2020, and for the vast majority (90%), this was in hazardous circumstances (adult had consumed alcohol, smoked, slept on a sofa, or the infant was premature or low birth weight and less than 3 months old). More deaths also occurred when there was a disruption in infant care routine compared with previous observations (52.6% vs 20.7%, p<0.001).

**Conclusions** A more targeted approach is needed with vulnerable families emphasising the importance of sleeping infants on their back and proactive planning infant sleep when there are disruptions to the normal routine, in particular to avoid hazardous co-sleeping.

## INTRODUCTION

Sudden infant death syndrome (SIDS) is defined as the unexpected death of a baby

### STRENGTHS AND LIMITATIONS OF THIS STUDY

⇒ Data from the case–control studies were collected prospectively with face-to-face interviews shortly after the death or reference sleep and using the same questions, so the data were easier to combine.

⇒ Using age-matched surviving controls in these studies provided us with a baseline from which we can establish associated background characteristics and risk factors.

⇒ Given the welcome reduction in sudden infant death syndrome deaths, national data provide us with sufficient numbers to observe patterns among the more recent deaths.

⇒ Only three quarters of all unexpected deaths from 2020 were fully reviewed and available for analysis.

⇒ The lack of recent control data, apart from what is available from national statistics, is a limiting factor.

aged under 1 year, which remains unexplained after investigation, including a post-mortem, thorough investigation of the death scene and circumstances of death and review of the clinical history.[1] In England and Wales, identification of risk factors in the infant sleep environment and subsequent uptake of safer sleep advice by caregivers has led to an almost 90% fall in these deaths over the last 35 years.[2]

With the welcome reduction in SIDS, the demographic profile has changed, and these tragedies are largely the preserve of families living in socially deprived conditions.[3] The population-wide risk reduction campaigns have been successful but if the modifiable risks previously identified are still occurring among current deaths, this would suggest that a more targeted approach is needed.

We conducted two prospective case–control observational studies in 1993–1996 and 2003–2006 in England after the 'Back to Sleep' campaign in 1991. The major risks identified

included infant sleeping position, hazardous sleep locations and social disruption in terms of changes in infant care routine.[3–6] The only subsequent observational case–control study was conducted in New Zealand in 2012–2015, which showed the risk surrounding hazardous bed-sharing was still present.[7]

The National Child Mortality Database (NCMD) was established in 2019, with the core aim to study and analyse the patterns, causes and associated risk factors of child mortality in England. The programme is commissioned by the Healthcare Quality Improvement Partnership (HQIP) as part of the National Clinical Audit and Patient Outcomes Programme (NCAPOP). HQIP is led by a consortium of the Academy of Medical Royal Colleges, the Royal College of Nursing and National Voices. NCAPOP is funded by NHS England, the Welsh Government and, with some individual projects, other devolved administrations and crown dependencies.

Notification of deaths to NCMD is required by statute within 48 hours. The database is co-ordinated at the University of Bristol, with locality-based Child Death Review Teams mandated since 2008[8] to report child deaths. A recent review of the circumstances and supplementary data on the major risk factors for unexplained infant deaths in England has been published.[9] Given the lengthy time it can take to conduct these investigations, a cut-off date for completed cases of June 2022 was chosen to report on 70% of unexplained deaths that occurred in 2020, The cohort used in this paper uses a cut-off date of September 2022 when 75% of deaths that occurred in 2020 had been reviewed.

This paper presents the data from unexplained infant deaths in England in 2020 reported to NCMD, alongside a combined analysis of our two earlier case–control studies to assess the relevance of previously identified risk factors and where increased emphasis might be required in risk reduction messaging if we were to adopt a more targeted approach.

## METHODS

Unexpected and unexplained infant deaths are derived by exclusion; by failing to demonstrate an adequate cause of death. The categorisation as an unexplained death in England is reached using a protocol that includes a detailed clinical history, review of the scene and circumstances of the death (ideally by a health professional together with a police officer), a standardised post-mortem examination by a paediatric pathologist (sometimes together with a forensic pathologist), additional investigations as appropriate and a final multi-agency child death review meeting.[9]

The outcome of the child death review meeting is further scrutinised, after anonymisation, by the local Child Death Overview Panel (CDOP), which is responsible for overseeing all investigations into child deaths in their area. There are 58 CDOPs in England.

### Data sources
#### The NCMD 2020 cohort of unexplained infant deaths
We report on the unexpected deaths of infants for which there was no immediately apparent cause of death (eg, trauma) that occurred between 1 January 2020 and 31 December 2020 that remained unexplained after investigations and the CDOP review had taken place. It is important to note that not all deaths in 2020 had a completed CDOP review (our cut-off was September 2022), which means numbers presented for 2020 will be underestimated. The NCMD has been collecting data since 2019, but the year 2020 was chosen to allow for the majority of deaths to have undergone review, and for newer data collection forms (introduced in 2019) to become established. The supplementary reporting form: sudden unexpected deaths[10] is required for eligible deaths and contains the majority of questions relating to the infant sleep environment.

#### The CESDI and SWISS combined data set
The Confidential Enquiry into Stillbirths and Deaths in Infancy (CESDI, 1993–1996) conducted in five English regions (Northern, South West, Trent, Wessex and Yorkshire) and the South West Infant Sleep Scene (SWISS, 2003–2006) Study were population-based case–control studies collecting data on all sudden and unexpected deaths in infancy over a set timeframe. Very few deaths were missed in the study areas. Both studies used similar methods with corresponding questions and categories for responses allowing for variables to be combined. Both studies used surviving control infants from the same population matched or weighted for age. A period of sleep (the 'reference sleep') was identified in the control infant's life in the 24 hours before the interview, corresponding to the time of day during which the index baby had died. Guidelines for strengthening the reporting of observational studies in epidemiology were followed.[11] Details of these studies have been published elsewhere.[4–6 12]

Exposure variables were similar across all data sets, with only notable differences for ethnicity (which was collected from the parents in the earlier case–control studies and recorded for the infant in the NCMD data) and socio-economic status (which used occupational classification in the earlier case–control studies, and Index of Multiple Deprivation score in the NCMD data). Wording of questions relating to the sleep environment were comparable across all three datasets.

### Variables definitions
#### Infant sleeping position
The sleeping position of the infant was a multicategorical variable with infants being placed supine (on the back), on their side or prone (front). Both the position found and put down were recorded and analysed.

#### Sleep location
A cot-type bed was defined as a cot, crib, Moses basket, three-sided baby bed attached to adult bed or a travel cot.

A baby carrier was defined as a pram, pushchair, bouncy chair or sling. An adult bed was either a standard UK single or double-sized bed.

### Co-sleeping

Co-sleeping was defined as an infant who was found sleeping next to an adult whether this was in an adult bed, on a sofa or in a chair. We have previously analysed the combined data to look specifically at co-sleeping[13] and confirmed significant interactions with hazardous practices (co-sleeping on a sofa or chair, co-sleeping next to an adult who has consumed more than two units of alcohol and co-sleeping next to an adult who habitually smokes).[12] The different hazards associated with co-sleeping were not mutually exclusive; some infants, for instance, slept on a sofa with an adult who smoked and had consumed alcohol. As with previous papers, a hierarchical approach of categorisation of co-sleeping was adopted based on the strength of risk reported from the two studies to ease interpretation of the findings. Thus, sofa-sharing, quantified as the highest risk from our previous studies, was categorised regardless of whether the parents consumed alcohol or smoked. Bed-sharing and alcohol consumption of two or more units were categorised regardless of smoking status. The remaining category of hazardous bed-sharing were infants who slept next to someone who smokes, or where the infant was vulnerable (defined as infants who were pre-term or low birthweight and <90 days old). Thus, the final category, representing bed-sharing in the absence of known hazards, was those parents who did not co-sleep on a sofa, had not consumed two or more units of alcohol, did not smoke, or sleep next to vulnerable infants.

### Change in routine

This is defined as a change in routine that may affect infant care for the last sleep; temporary accommodation, family on holiday, family visit to friends or relatives, party or celebration, having overnight guest(s) stay, taking long journeys, etc.

### Statistical methodology

For ease of interpretation, the variables were dichotomised using standard definitions, if available, or previous definitions used in the earlier studies. All denominators were provided when quoting prevalence to indicate where data were missing. $\chi^2$ was calculated using Pearson's method, if any expected cell frequencies were less than 5, Fisher's exact test was used. Continuous data that were not normally distributed were tested using the Mann-Whitney U test and described with medians and IQR. P values are reported with a significance threshold of .05.

### Patient and public involvement

Families remain at the heart of our research. Our Family Advisory Group has been instrumental in advising us on how to interpret the findings of this analysis, by supporting the development of tools and resources to support families with following safer sleep advice, as part of our wider work.[14]

## RESULTS

### Ascertainment

Of the 361 sudden and unexpected infant deaths that occurred in England during 2020, 75% (n=270) had been reviewed by a CDOP by September 2022. Of these, 138 were classified as unexpected and unexplained infant deaths (ie, SIDS), with 133/138 (96.4%) thought to occur during sleep. Of the 405 SIDS infants and 1387 controls in the two case–control studies, data were available on the sleep environment in which the infant was found for 402 SIDS infants (99.3%) and 1387 controls (100%).

### Background characteristics

The background characteristics of the SIDS infants and families that distinguished them from surviving controls in the previous observational studies were just as apparent if not more so among the more recent deaths in 2020 (table 1). Male preponderance and births to multiparous mothers remained a feature of SIDS while infant vulnerability has become more marked. The median age of SIDS deaths is now nearer 2 months than 3 months old. Low birth weight, preterm birth, multiple births and admission to a neonatal unit are recognised significant characteristics of SIDS infants compared with age-matched surviving controls, but the prevalence for each of these factors was greater in the more recent SIDS deaths, with 36.2% of the SIDS deaths in 2020 exhibiting at least one of these characteristics. The prevalence of maternal smoking during pregnancy among the SIDS deaths seems to have fallen from 65% to 52.1%, but the prevalence of maternal smoking at delivery in England has also reduced in this time period by a third from 15.8% in 2006 to 9.6% in 2020,[15] suggesting a 5–6-fold greater prevalence of maternal smokers among current SIDS deaths. The median maternal age among the SIDS deaths has increased from 24 years in previous studies to 27 years in 2020, but in the same time period, the average age for first-time mothers in England & Wales has increased from 27.5 years in 2007 to 29.1 years in 2020.[16] Ethnicity was difficult to measure because the status in the observational studies was taken from the parents, while in the NCMD data, it was recorded for the infant, thus robust estimates are only available for those categorised as 'white'. In 2020, this category was prevalent in three quarters of the SIDS deaths, 12.7% of infants were of mixed ethnic origin, 7.5% were 'Black or Black British' and 6% were 'Asian or Asian British'. It was also difficult to compare socioeconomic status as different metrics were used. In the NCMD data, 41.3% of the SIDS families were in the most deprived quintile as measured by the deprivation index based on postcode. Socioeconomic status in the observational studies was measured using occupational classification, 17.4% of the control families were classified as semiskilled, unskilled or unemployed compared with

**Table 1**  Comparison of infant and family characteristics (combined study data vs NCMD)

| Variable | Category | Data from observational studies | | | | Data from NCMD | | 1) vs 2)* |
| | | 1) SIDS | | Surviving controls | | 2) SIDS | | P value |
| | | n/N | % | n/N | % | n/N | % | |
| Gender | Male | 254/402 | 63.2 | 728/1387 | 52.5 | 86/138 | 62.3 | 0.85 |
| Birth weight | <2500 g | 87/402 | 21.6 | 68/1379 | 4.9 | 39/128 | 30.5 | 0.04 |
| Parity | >1 child | 284/402 | 71.1 | 794/1387 | 57.2 | 87/126 | 69.1 | 0.67 |
| Gestational age | Pre-term (<37 weeks) | 77/400 | 19.3 | 72/1375 | 5.2 | 40/135 | 29.6 | 0.012 |
| Multiple birth | Twin or triplet | 18/402 | 4.5 | 13/1387 | 0.9 | 11/137 | 8.0 | 0.12 |
| Admission to NICU | At birth | 94/400 | 23.5 | 99/1378 | 7.2 | 39/136 | 28.7 | 0.23 |
| Maternal smoking | During pregnancy | 258/398 | 64.8 | 360/1386 | 26.0 | 62/119 | 52.1 | 0.012 |
| Ethnicity | White† | 341/389 | 87.7 | 1164/1378 | 91.7 | 97/130 | 74.6 | <0.001 |
| | | **Median** | **IQR** | **Median** | **IQR** | **Median** | **IQR** | |
| Infant age | At death/ref sleep | 86 days | 52–148 | 100 days | 62–162 | 66 days | 34–118 | 0.003‡ |
| Maternal age | At death/ref sleep | 24 years | (21–29) | 28 years | 24–31 | 27 years | 22–32 | <0.001‡ |

*Comparing prevalence of SIDS from the observational studies with SIDS from NCMD 2020.
†Infant ethnicity was not recorded in the observational studies, white ethnicity was derived from ethnicity of mother and biological father but no further breakdown in ethnicity grouping could be made.
‡Mann-Whitney U test.
NCMD, National Child Mortality Database; NICU, neonatal intensive care unit; SIDS, sudden infant death syndrome.

44.1% of the SIDS families. Thus, marked socioeconomic deprivation among SIDS deaths (however measured) seems to have been maintained.

### Factors in the infant sleeping environment

Among the recent SIDS deaths, more infants were placed on their back for the last sleep and fewer on their side compared with previous observations, but the same proportion was placed on their front; the proportion found on their front had increased (table 2). In the NCMD cohort, significantly fewer of the SIDS infants were found in a cot-type bed and significantly more in an adult bed either sharing the sleeping surface or sleeping alone. More than half of the SIDS infants in 2020 were sleeping next to an adult and 63/70 (90%) of these in hazardous conditions; co-sleeping on a sofa or chair, next to an adult who had consumed alcohol or smoked or an adult sleeping next to a young infant (<3 months old) who was born prematurely (<37 weeks) or with low birth weight (<2500 g). Illegal drug consumption (mainly cannabis and opioids) prior to the last sleep was not collected in the CESDI study but was collected in the SWISS study and 13% of the SIDS infants slept next to an adult who had consumed these drugs compared with 3.4% of the controls. In 2020 of those families who answered this question, 11/50 (22%) had consumed drugs prior to sleeping with the infant. In previous studies, a change in infant care routine for the last sleep was a significant predictor

and this was even more marked in recent deaths; occurring in 53% of the SIDS deaths in 2020.

### DISCUSSION

Despite three decades of advising parents to place young infants on their back for sleep in England, the proportion put down and found prone remains high among SIDS deaths. Of those found prone half were less than 4 months old, an age, in terms of infant physiology, where it is particularly difficult for infants to extricate themselves from this position. More than half the SIDS deaths in 2020 occurred when the infant was sharing a sleep surface with an adult, predominantly in hazardous circumstances. Current advice in the UK acknowledges that bed-sharing supports breastfeeding[17] and can happen both in planned and unplanned ways.[18] Health professionals are encouraged to describe the circumstances, which make bed-sharing more risky, namely in the presence of alcohol, smoking or with a vulnerable (low birth weight or premature) infant.[13] They also include advice to avoid bed-sharing where there has been any drug use or medications that can impair arousability, and our findings suggest this needs to be emphasised. Bed-sharing with infants less than 3 months old who were born prematurely or with low birth weight also needs to be emphasised along with not leaving infants to sleep alone in an adult bed. The finding that very low numbers of deaths

**Table 2** Comparison of risk factors in the sleeping environment (combined study data vs NCMD)

| Variable | Category | Data from observational studies | | | | Data from NCMD | | 1) vs 2)* |
| | | 1) SIDS | | Surviving controls | | 2) SIDS | | P value |
| | | n/N | % | n/N | % | n/N | % | |
| Sleeping position | Supine | 191/391 | 48.8 | 968/1382 | 70.0 | 61/90 | 67.8 | 0.001 |
| Put down | Side | 143/391 | 36.6 | 370/1382 | 26.8 | 15/90 | 16.7 | <0.001 |
| | Prone | 57/391 | 14.6 | 44/1382 | 3.2 | 14/90 | 15.6 | 0.81 |
| Sleeping position | Supine | 159/382 | 41.6 | 1106/1333 | 83.0 | 47/89 | 52.8 | 0.06 |
| Found | Side | 88/382 | 23.0 | 141/1333 | 10.6 | 6/89 | 6.7 | <0.001 |
| | Prone | 135/382 | 35.3 | 86/1333 | 6.5 | 36/89 | 40.4 | 0.37 |
| Sleeping location | Cot-type bed† | 202/395 | 51.1 | 1021/1383 | 73.8 | 42/132 | 31.8 | <0.001 |
| | Baby carrier‡ | 22/395 | 5.6 | 94/1383 | 6.8 | 4/132 | 3.0 | 0.24 |
| | Car seat | 3/395 | 0.8 | 14/1383 | 1.0 | 1/132 | 0.8 | 1.0¶¶ |
| | Adult bed (sharing) | 112/395 | 28.4 | 206/1383 | 14.9 | 55/132 | 41.7 | 0.004 |
| | Adult bed (alone) | 16/395 | 4.1 | 7/1383 | 0.5 | 12/132 | 9.1 | 0.03 |
| | Sofa/chair (sharing) | 33/395 | 8.4 | 7/1383 | 0.5 | 15/132 | 11.4 | 0.30 |
| | Sofa/chair (alone) | 6/395 | 1.5 | 28/1383 | 2.0 | 1/132 | 0.8 | 0.69¶¶ |
| | Other§ | 1/395 | 0.3 | 6/1383 | 0.4 | 2/132 | 1.5 | 0.16¶¶ |
| Co-sleeping | Not co-sleeping | 252/397 | 63.5 | 1174/1386 | 84.7 | 62/132 | 47.0 | <0.001 |
| | Any co-sleeping | 145/397 | 36.5 | 212/1386 | 15.3 | 70/132 | 53.0 | |
| For the last sleep | Hazardous co-sleeping¶: | | | | | | | |
| | On a sofa/chair | 33/397 | 8.3 | 7/1386 | 0.5 | 15/132 | 11.4 | 0.29 |
| | Alcohol consumed | 29/397 | 7.3 | 12/1386 | 0.9 | 18**/132 | 13.6 | 0.03 |
| | Smoking or vulnerable†† | 65/397 | 16.4 | 66/1386 | 4.8 | 30/132 | 22.7 | 0.10 |
| | Co-sleeping no hazards | 18/397 | 4.5 | 127/1386 | 9.2 | 7‡‡/132 | 5.3 | 0.72 |
| Change in routine | For the last sleep§§ | 81/391 | 20.7 | 171/1383 | 12.4 | 30/57 | 52.6 | <0.001 |

*Comparing prevalence of SIDS from the observational study with SIDS from NCMD 2020.
†Includes, cot, crib, Moses basket, three-sided baby bed attached to adult bed or a travel cot.
‡Includes pram, pushchair, bouncy chair or sling.
§ Includes on floor, in an electronic settling device, waterbed, pod, nest or bean bag.
¶Consumption of illegal drugs (cannabis/opioids) prior to co-sleeping was not recorded in the CESDI study. In the SWISS study, 10/77 cases (13.0%) slept next to an adult who had taken illegal drugs compared with 3/87 of the controls (3.4%). In the NCMD cohort, it was 11/50 (22.0%) of those who were asked this question.
**This question was only asked for 50 of the 70 co-sleepers so will be an underestimate.
††Adult co-sleeper smoked or infant was vulnerable (infants who were pre-term or low birth weight and <90 days old).
‡‡One infant co-slept with another child, one infant slept next to an adult who had taken cannabis and for the remaining five infants the question on alcohol consumption was not asked.
§§Change in routine involving the infant (eg, on holiday, visit to friends or relatives, party, overnight guest(s), long journey, etc).
¶¶Using Fisher's Exact test.
NCMD, National Child Mortality Database; SIDS, sudden infant death syndrome.

occur outside of these known risk factors supports the current continuation of the advice to emphasise when not to bed-share.

We have previously reported major changes in the epidemiology of SIDS in 2006.[19] Between 1984 and 2003 in Avon, the proportion of SIDS victims from socioeconomically deprived families, vulnerable infants (low birth weight, preterm, multiple births) and infants exposed to tobacco smoke increased over time. During this 20-year period, the national SIDS rate fell almost 10-fold from 2.3 deaths to 0.26 per 1000 livebirths; the number of co-sleeping deaths also halved across the two decades but proportionally increased from 12% of all SIDS deaths to 50%. Our case–control study conducted in 2003–2006 reported that 54% of the SIDS deaths occurred while sleeping next to an adult and the median age of SIDS had fallen to 66 days. Recent findings from the NCMD in 2020 have confirmed these earlier findings regarding bed-sharing and infant age and that social deprivation remains a key feature of overall infant and child mortality.[20] Observations of all unexpected infant deaths in 2020 suggest that deprivation is still a strong characteristic and the prevalence of infant vulnerability has risen; over a third of the current deaths were either low birth weight, preterm or a multiple birth. Birth characteristics from England in 2020[16] suggest low birth weight (6.5%), preterm births (7.4%) and multiple births (1.4%) have all slightly increased in the population compared with the previous study control data, but the prevalence among the SIDS deaths in 2020 for these three characteristics was 4–5-fold higher. The prevalence of maternal smoking has fallen from 65% in the previous studies to 52% among the recent SIDS deaths but is still extremely high, fivefold higher than the levels of maternal smoking at delivery nationally[15] in England; suggesting this is still a strong predictor of these deaths. Data from 2020 in England and Wales also suggest 15.7% of births were to mothers aged <25 years[16] compared with 46/131 (35.1%) among the SIDS mothers in the NCMD cohort suggesting young maternal age is still a factor.

The persistence of SIDS risk factors among current deaths despite national risk reduction campaigns is not unique to the UK. In the Netherlands,[21] Spain,[22] Germany[23] and Australia,[24] the proportion of infants placed in a non-supine position is increasing while a survey in Norway[25] suggests hazardous bed-sharing is far more common than initially anticipated. In a case–control study conducted in New Zealand in 2012–2015, hazardous bed-sharing especially the interaction with maternal smoking was cited as the greatest risk for unexpected infant deaths.[26]

## Strengths and limitations

Given the prospective nature of the case–control studies, the detailed data collected shortly after the death or reference sleep, using face to face interviews and using the same questions, so the data were easier to combine can be seen as strengths of our investigations. The

deaths were also carefully classified using multidisciplinary panels for both studies and the recent cohort. Only 75% of the deaths in 2020 had been reviewed by the chosen cut-off date (partially due to delays caused by the pandemic) and lack of recent control data can be seen as limitations although routinely collected characteristics surrounding the birth at the national-level alleviates some of this problem. Socioeconomic deprivation was measured differently in the studies compared with the recent cohort, but both measures indicate a significant proportion of SIDS families are in the most deprived grouping. Observing changes in different ethnic groups was limited.

## Recommendations

The NCMD is a unique resource that can monitor the known risks in the infant sleep environment on an annual basis and could provide evidence of where prevention efforts should focus. Certainly, the finding that almost half of the deaths occur in hazardous co-sleeping circumstances should be a call to all those involved in improving the uptake of safer sleep advice to find ways to support families to reduce the risks for their infants from these situations. Sofa sharing and bed-sharing in the presence of an adult who smokes, has consumed alcohol or taken drugs should be the first priorities. The finding that changes to the infant care routine are increasingly prevalent in deaths, and that combinations of risk factors make the situation worse,[27] suggests that prevention efforts may also benefit from interventions that directly focus on planning for safety during times of disruption to the normal routine. Our work with vulnerable families has shown that parents appreciate individually tailored advice that provides reasons, giving the how and why certain situations can increase risks for a baby.[28–30] The increased vulnerability exhibited among current SIDS deaths and continued exposure to several modifiable risk factors within the infant sleep environment despite widespread population-based risk reduction campaigns, suggest targeting resources to families most in need may reduce deaths further. Our specific recommendations based on these findings, are:

1. National SIDS risk reduction campaigns should focus on renewed efforts to emphasise the need for all babies to be put down for sleep on their backs and avoid hazardous co-sleeping; in particular, to avoid using sofas or consuming alcohol or drugs before bed-sharing.
2. Families with infants at increased risk should be provided with targeted intensive support, particularly in the first few months of a baby's life, to address the barriers to following safer sleep advice, and plan for infant sleep safety during times of disruption to the normal routine.
3. Annual monitoring by CDOPs and the NCMD should include reporting on background characteristics of families affected by unexpected infant deaths, track sleep environment risk factors present in the deaths each year and provide geographical 'heat maps',

showing areas that may need more intensive intervention and support.

**Contributors** AP acts as guarantor and accepts full responsibility for the work and/or the conduct of the study, had access to the data, and controlled the decision to publish. NT contributed to analysis and interpretation of data, drafting and review of the manuscript. JI participated in the study concept and design, contributed to interpretation of data, drafting and review of the manuscript. PF participated in the study concept and design, contributed to data acquisition, interpretation of data, drafting and review of the manuscript. KP participated in the study concept and design, contributed to interpretation of data, drafting and review of the manuscript. TW participated in the study concept and design, contributed to data acquisition, interpretation of data, drafting and review of the manuscript. VS participated in the study concept and design, contributed to data acquisition, interpretation of data, drafting and review of the manuscript. KP participated in the study concept and design, contributed to interpretation of data, drafting and review of the manuscript. KL participated in the study concept and design, contributed to data acquisition, interpretation of data, drafting and review of the manuscript. BA participated in the study concept and design, contributed to interpretation of data, drafting and review of the manuscript. PB participated in the study concept and design, contributed to data acquisition, analysis, interpretation of data, drafting and review of the manuscript. All authors have seen and approved the final version of the manuscript.

**Funding** Dr Anna Pease, Research Fellow, NIHR300820, is funded by the NIHR for this research project. The views expressed in this publication are those of the authors and not necessarily those of the NIHR, NHS or the UK Department of Health and Social Care. The National Child Mortality Database (NCMD) Programme is commissioned by the Healthcare Quality Improvement Partnership (HQIP) as part of the National Clinical Audit and Patient Outcomes Programme (NCAPOP). HQIP is led by a consortium of the Academy of Medical Royal Colleges, the Royal College of Nursing, and National Voices. Its aim is to promote quality improvement in patient outcomes. HQIP holds the contract to commission, manage and develop the National Clinical Audit and Patient Outcomes Programme (NCAPOP), comprising around 40 projects covering care provided to people with a wide range of medical, surgical and mental health conditions. NCAPOP is funded by NHS England, the Welsh Government and, with some individual projects, other devolved administrations and crown dependencies www.hqip.org.uk/national-programmes. NHS England provided additional funding to the NCMD to enable rapid set-up of the real-time surveillance system and staff time to support its function but had no input into the data analysis or interpretation.

**Competing interests** None declared.

**Patient and public involvement** Patients and/or the public were involved in the design, or conduct, or reporting, or dissemination plans of this research. Refer to the Methods section for further details.

**Patient consent for publication** Consent obtained directly from patient(s).

**Ethics approval** Approval was sought and gained from each regional research ethics committee and by each constituent local research ethics committee for the two case-control studies.The NCMD legal basis to collect confidential and personal level data under the Common Law Duty of Confidentiality has been established through the Children Act 2004 Sections M-N, Working Together to Safeguard Children 2018 (https://consult.education.gov.uk/child-protection-safeguarding-and-family-law/working-together-to-safeguard-children-revisions-t/supporting_documents/WorkingTogethertoSafeguardChildren.pdf) and associated Child Death Review Statutory & Operational Guidance https://assets.publishing.service.gov.uk/government/uploads/system/uploads/attachment_data/file/859302/child-death-review-statutory-and-operational-guidance-england.pdf).

**Provenance and peer review** Not commissioned; externally peer reviewed.

**Data availability statement** Data are available upon reasonable request. Data are available upon reasonable request. Aggregate data may be available on request to the National Child Mortality Database, and subject to approval by HQIP.

**ORCID iDs**
Anna Pease http://orcid.org/0000-0002-3472-1047
Nicholas Turner http://orcid.org/0000-0003-1591-6997
Jenny Ingram http://orcid.org/0000-0003-2366-008X
Kieren Pitts http://orcid.org/0000-0002-0927-7677
Karen Luyt http://orcid.org/0000-0002-9806-1092
Becky Ali http://orcid.org/0000-0002-8991-9616
Peter Blair http://orcid.org/0000-0002-7832-8087

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
