## [Reviewer comments · BMJ Open]

ARTICLE DETAILS

TITLE (PROVISIONAL)	Changes in background characteristics and risk factors amongst SIDS infants in England: cohort comparisons from 1993 to 2020
AUTHORS	Pease, Anna; Turner, Nicholas; Ingram, Jenny; Fleming, Peter; Patrick, Karen; Williams, Tom; Sleep, Vicky; Pitts, Kieren; Luyt, Karen; Ali, Becky; Blair, Peter

VERSION 1 – REVIEW

REVIEWER	Osei-Poku, Godwin K. Boston University School of Public Health
REVIEW RETURNED	13-Jul-2023

GENERAL COMMENTS	Thank you for the opportunity to review this article. This is well-written and answers an important question. Despite the reduction in SIDS in the last decade, a lot more education on risk reduction strategies is needed to reduce SIDS rates especially in socially disadvantaged communities. The article attempts to identify areas for targeted interventions by comparing the recent prevalence of SIDS risk factors with historical estimates from 1993-1996 and 2003-2006. While the article succeeded in making this comparison, I have a few suggestions to strengthen the paper. Abstract: Line 27-28, page 2, admission to NICU does not appear to be significant with p-value of 0.23, hence saying that infants who died in 2020 had an increased prevalence of admission to a Neonatal Intensive Care Unit (28.7% vs 23.5%, $P=0.23$) may not be entirely accurate since the p-value of 0.23 is greater than 0.05 (assuming that is the alpha for significance). Please consider rephrasing this.Lines 30-34, page 2: It might be better to rephrase this to highlight that there was no change in the proportion of infants who were put down or found prone between the recent SIDS cases and the historical comparator. Perhaps consider “Among recent deaths, fewer infants were put down or found on their side, however, there was no significant change in the proportion of infants who were put down (15.6% vs 14.6%, $p=0.81$) and found prone (40.4% vs 35.3%, $p=0.37$), despite population wide risk-reduction advice over three decades”. The way it is framed is not very clear and presenting non-significant p values this way may be confusing to a reader. Methods: The authors indicate that the CESDI and SWISS studies “used similar methods and corresponding questions and categories for responses allowing for variables to be combined”. However, it is not stated or clear if the NCMD used similar questions or categories for the 2020 data collection. Can the authors clarify if the data collection instruments were similar
---

	across all studies to allow a like to like comparison of findings. For example, were the newer data collection forms introduced in 2019 similar to those used in the prior studies? Perhaps, a statement or two in the methods clarifying which exposure variables were similar or different across the case-control cohort and the NCMD cohort would be better. Line 33-34 and 44-45 on page 8 (results) highlight some of the differences in definitions but I feel a section on exposure ascertainment in the methods might be better. 2. Statistical methodology: Please indicate the significance level used so readers can interpret the p-values. Results: 1. Some sections of the results would be better in the discussion. For instance, Lines 12-25, page 8 would be better placed in the discussion since it interprets the results Discussion: Overall, the discussion ties in the findings and the recommendations are sound. I agree that targeted interventions are needed specifically for communities at risk of SIDS, especially those in socially disadvantaged communities.
--	--

REVIEWER	Greenough, Anne King's College London, Department of Women and Children's Health, School of Life Sciences, Faculty of Life Science and Medicine
REVIEW RETURNED	17-Jul-2023

GENERAL COMMENTS	The National Child Mortality database has been used to investigate background characteristics and risk factors in the sleeping environment associated with SIDS in 2020 and compared the prevalence with previous English SIDS case control studies. The authors demonstrate in SIDS cases an increased prevalence of LBW and preterm births and an excess of socio-economically deprived families, male infants and high levels of smoking during pregnancy. More deaths occurred also when there had been a disruption in infant care routine compared to other studies. The authors conclude a more targeted approach is needed with vulnerable families. Comments 1. This is a well done study using national data emphasizing important risk factors for SIDS in 2020. 2. The limitations are well described including the lack of full ethnicity data in the mothers which would be important for fully targeting interventions. 3. The implication in the abstract is that admission to a NICU was a risk factor but this does not reach statistical significance and should be removed.
--

VERSION 1 – AUTHOR RESPONSE

Reviewer 1	
Line 27-28, page 2, admission to NICU does not appear to be significant with p-value of 0.23, hence saying that infants who died in 2020 had an increased prevalence of admission to a Neonatal Intensive Care Unit (28.7% vs 23.5%, P=0.23) may not be entirely	Thank you – this was an error and we have removed this finding from the abstract, as requested by reviewer 2.

accurate since the pvalue of 0.23 is greater than 0.05 (assuming that is the alpha for significance). Please consider rephrasing this.	
Lines 30-34, page 2: It might be better to rephrase this to highlight that there was no change in the proportion of infants who were put down or found prone between the recent SIDS cases and the historical comparator. Perhaps consider “Among recent deaths, fewer infants were put down or found on their side, however, there was no significant change in the proportion of infants who were put down (15.6% vs 14.6%, p=0.81) and found prone (40.4% vs 35.3%, p=0.37), despite population wide risk-reduction advice over three decades”. The way it is framed is not very clear and presenting non-significant p values this way may be confusing to a reader.	Thank you, we agree and have changed this wording as suggested.
The authors indicate that the CESDI and SWISS studies “used similar methods and corresponding questions and categories for responses allowing for variables to be combined”. However, it is not stated or clear if the NCMD used similar questions or categories for the 2020 data collection. Can the authors clarify if the data collection instruments were similar across all studies to allow a like to like comparison of findings. For example, were the newer data collection forms introduced in 2019 similar to those used in the prior studies? Perhaps, a statement or two in the methods clarifying which exposure variables were similar or different across the case-control cohort and the NCMD cohort would be better. Line 33-34 and 44-45 on page 8 (results) highlight some of the differences in definitions but I feel a section on exposure ascertainment in the methods might be better.	Thank you, we have included the following section in the methods: “Exposure variables were similar across all datasets, with only notable differences for ethnicity (which was collected from the parents in the earlier case control studies, and recorded for the infant in the NCMD data), and socio-economic status (which used occupational classification in the earlier case control studies, and Index of Multiple Deprivation score in the NCMD data). Wording of questions relating to the sleep environment were comparable across all three datasets.”
Statistical methodology: Please indicate the significance level used so readers can interpret the p-values.	Thank you we have added the following to the methods: “P values are reported with a significance threshold of .05.”
Some sections of the results would be better in the discussion. For instance, Lines 12-25, page 8 would be better placed in the discussion since it interprets the results	Thank you, we have moved the following sections into the discussion: “Birth characteristics from England in 2020 (15) suggest low birthweight (6.5%), pre-term births (7.4%) and multiple births (1.4%) have all slightly increased in the population compared to the previous study control data but the prevalence amongst the SIDS deaths in 2020 for these three characteristics was 4-5 fold higher.” “Data from 2020 in England & Wales also suggest 15.7% of births were to mothers aged <25 years (15) compared to 46/131 (35.1%) among the SIDS mothers in the NCMD

	cohort suggesting young maternal age is still a factor.”
Reviewer 2	
The implication in the abstract is that admission to a NICU was a risk factor but this does not reach statistical significance and should be removed.	Thank you – this was an error and we have removed this finding from the abstract, as requested.

VERSION 2 – REVIEW

REVIEWER	Osei-Poku, Godwin K. Boston University School of Public Health
REVIEW RETURNED	14-Aug-2023
GENERAL COMMENTS	Thank you for addressing my comments

VERSION 2 – AUTHOR RESPONSE